# Plant nutation relies on steady propagation of spatially asymmetric growth pattern

Mathieu Rivière[1,2] , Alexis Peaucelle[1,3], Julien Derr[1,4] and Stéphane Douady[1]

[1]Université Paris Cité, CNRS, Matière et Systèmes Complexes, F-75013 Paris, France; [2]Aix Marseille Univ, CNRS, IUSTI, Marseille, France; [3]Université Paris-Saclay, INRAE, AgroParisTech, Institut Jean-Pierre Bourgin, 78000, Versailles, France; [4]ENS de Lyon, Univ Lyon, CNRS, INRAE, Inria, Laboratoire de Reproduction et Développement des Plantes, Lyon, France

## Original Research Article

**Keywords:**
morphogenetic motions; growth kinematics; plant nutation; leaf development.

**Corresponding author:**
Julien Derr;
Email: julien.derr@ens-lyon.fr

**Associate Editor:**
Dr. Félix Hartmann

## Abstract

Nutation is one of the most striking and ubiquitous examples of the rhythmic nature of plant development. Although the consensus is that this wide oscillatory motion is driven by growth, its internal mechanisms remain to be fully elucidated. In this work, we study the specific case of nutation in compound leaves of the *Averrhoa carambola* plant. We quantify the macroscopic growth kinematics with time lapse imaging, image analysis and modelling. Our results highlight a distinct spatial region along the rachis—situated between the growth and mature zones—where the differential growth driving nutation is localised. This region coincides with the basal edge of the growth zone, where the average growth rate drops. We further show that this specific spatiotemporal growth pattern implies localised contraction events within the plant tissue.

## 1. Introduction

Plants move. This overlooked truth has come to light again thanks to the recent study of spectacular ultra-fast motions (Forterre et al., 2016). For example, the snapping of the Venus flytrap (Forterre et al., 2005; Sachse et al., 2020) and the catapulting of fern spores (Noblin et al., 2012) both require high speed cameras to be recorded. At the opposite side of the timescales spectrum, plants move through their growth. The observation of these slow observation necessitates time-lapse imaging. After Darwin (Darwin, 1897), they started to be historically investigated with the development of photography (Gaycken, 2012). But we are still evidencing nowadays a variety of exciting new motions (Derr et al., 2018; Rivière et al., 2017, 2020). They can either be nastic motions, or tropisms, depending on whether the direction of the motion is imposed by factors internal or external to the plant, respectively. The movement is defined as autonomic (respectively, paratonic) depending on whether the triggering signal is internal to the plant or not. They can finally be reversible or linked to irreversible growth. These three dichotomies define the traditional classification of slow plant motions (Rivière et al., 2017). Within this framework, the status of one remarkable movement called nutation is still undecided (Baskin, 2015; Mugnai et al., 2015; Rivière et al., 2017; Stolarz, 2009)

Nutation is the phenomenon that causes the orientation of the long axis of an elongated growing plant to vary over time in a pseudo-periodical way. It was already observed for climbing plants by British botanists of the 17th century (Webster, 1966) and began to be studied by Hugo von Mohl and Ludwig Palm in the first part of the 19th century (Baillaud, 1957). To the best of our knowledge, the term 'nutation' was first mentioned by Charles Bonnet (Bonnet, 1754) although he acknowledges that this term had been named before him, by physicists who knew the phenomenon. They probably saw this motion as a botanical analog to the astronomical nutation.

Darwin later introduced the idea that nutation had an endogenous origin and many plant motions were actually modified nutations (Darwin, 1897). The very origin of nutation was a source of debate at the time nonetheless (Baillaud, 1957), and it remains so up to this date (Brown, 1993; Migliaccio et al., 2013; Mugnai et al., 2015). Part of the community backs up

Darwin's idea of an internal oscillator (Brown et al., 1990; Johnsson et al., 1999). Others ascribe this oscillating behaviour to inertial overshooting of the plant occurring during its straightening process (Agostinelli et al., 2020; Gradmann, 1922; Israelsson & Johnsson, 1967; Johnsson & Israelsson, 1968). Finally, the compromise solution calling for a combination of these two hypotheses gathers more and more support (Agostinelli et al., 2021; Johnsson, 1997; Johnsson et al., 1999; Orbović & Poff, 1997; Stolarz, 2009). The one thing making consensus is that nutation is a macroscopic manifestation of multicellular microscopic growth.

Plant growth results from a subtle balance between the strong internal osmotic pressure and the resisting rheology of the cell wall (Tomos et al., 1989). Although growth and plasticity are very distinct processes, and growth doesn't involve viscosity (Goriely, 2017), Lockhart used a Bingham plastic framework to formalise plant growth (Lockhart, 1965). Lockhart's model received good experimental support at the single cell level (Cosgrove, 1985; Green et al., 1971; Zhu & Boyer, 1992). Still, some shortcomings need to be addressed (Jordan & Dumais, 2010), and the origin of the cell wall-loosening mechanism remains unclear (Höfte et al., 2012; Kroeger et al., 2011; Micheli, 2001; Palin & Geitmann, 2012). The cell wall is considered here to be an inactive gel but it was demonstrated that elements of the cell wall, the homogalacturonans (HG) can transform chemical modification into mechanical expansion through cell controlled enzymatic demethylesterification (Haas et al., 2020). The precise role of elasticity that was added to Lockhart model later on by Ortega (Ortega, 1985) is then subject to debate (Haas et al., 2020; Kierzkowski et al., 2012). Finally, the multi-cellular aspect of the biophysics of growth remains to be understood (Boudon et al., 2015). In particular, dynamical aspects related to water fluxes between cells have just started to be taken into account, either numerically (Cheddadi et al., 2019) or even more recently theoretically with the development of a hydromechanical field theory for plant morphogenesis (Oliveri & Cheddadi, 2025). These new theoretical concepts will be key to understand the complex spatio-temporal behaviour observed in plant nutation.

The seminal work on the spatio-temporal characterization of nutation has been performed by Berg and Peacock (Berg & Peacock, 1992) where they evidenced strong fluctuations and traveling waves of the axial elongation rate in the sunflower hypocotyl. They even measured negative rates, suggesting local contractions. At the time, they acquired data with a single camera, and their growth measurements were necessarily biased by strong projection artifacts due to the three-dimensional nature of the motion.

Here, we aim to revisit in detail the phenomenon of nutation. By carefully quantifying the motion of nutation (taking into account the 3D nature of the motion), we will gain knowledge on the nature of this puzzling mechanism. In this article, we focus on the plant *Averrhoa carambola*, a plant known for exhibiting ample nutation (see Figure 1a, b) and other growth motions (Rivière, 2017; Rivière et al., 2017, 2020).

The manuscript is organised as follows. We start by characterising the kinematics of nutation at the scale of the whole leaf, and emphasise the spatial organisation of growth. Our measurements allow to characterise the growth law of nutation and highlight a relationship between mean growth and differential growth. We then zoom in on the bending zone and, thanks to a kinematics model, analyse contraction events. Finally, we put our results in perspective with the microscopic properties (elasticity and chemical content) of the plant cell wall.

## 2. Materials and methods

### 2.1. Growth conditions of plants

*Averrhoa carambola* seeds were directly obtained from commercially available fruits and sown into all-purpose compost. Young seedlings were first kept inside a small lab greenhouse. Older plants (> 6 months) were then moved to the experimentation room. There, plants were submitted to a 12/12 light cycle under ORTICA 200W 2700K culture lamps. The temperature and relative humidity rate were monitored with a DHT22 sensor. Temperature was usually comprised between $20^oC$ and $24^oC$. The relative humidity rate was around 60%. All methods were performed in accordance with the relevant guidelines and regulations.

### 2.2. Kinematics: sample preparation

The rachis of interest was carefully coated with fluorescent pigments with a brush. For curvature and coarse elongation measurements, the top of the rachis was coated homogeneously with orange pigments. Small blue fluorescent dots were added to mark the nodes and the petiole. For fine measurements of local growth, the orange pigments were deposited on the face of a few interfoliolar segments so that they form highly textured and contrasted patterns. In both cases, because of growth, pigments needed to be added manually on a regular basis to compensate the dilution of the signal over time.

### 2.3. Kinematics: image acquisition

The kinematics of nutation were captured using time-lapse photography with a DSLR camera controlled with the open-source software gPhoto2. The camera was firmly fixed to a rigid structure to avoid any displacement or rotation. The built-in flash of the camera was covered with LEE Moss green filter and set to the lowest intensity to keep light input minimal during nights. For curvature and coarse growth kinematics, top-views were taken every 2.5 min. For local growth measurements, side-views were taken every minute.

### 2.4. Kinematics: data analysis

The midline, or skeleton, of the rachis was obtained by first thresholding the red channel of the pictures. A cloud of points was obtained and then reduced to a smooth line with a moving median filter. The curvature of the rachis $\kappa_\perp$ in the plane of interest was obtained by locally fitting the midline to a circle. The position of the leaflets was retrieved by thresholding the blue channel. Because of growth, blues dots dilated, lost intensity in time and sometimes even split. The global unfurling motion of the rachis sometimes resulted in a temporary occlusion of some blue dots. Simple rules on the conservation of these dots, distance between consecutive dots and displacements values could overcome a majority of tracking failures. Manual correction was still needed in some special cases. Finally, the presented spatiotemporal graphs were smoothed with 2D averaging and median filters.

### 2.5. Kinematics: fine measurements

We obtained the skeleton of the rachis by a simple geometric transformation of the upper contour which is less altered by leaflet motions. We measured the elongation field along the rachis by using a previously published image-to-image correlation algorithm (Bastien et al., 2016). The time-frequency analysis of the elongation signals was done by using MATLAB's continuous

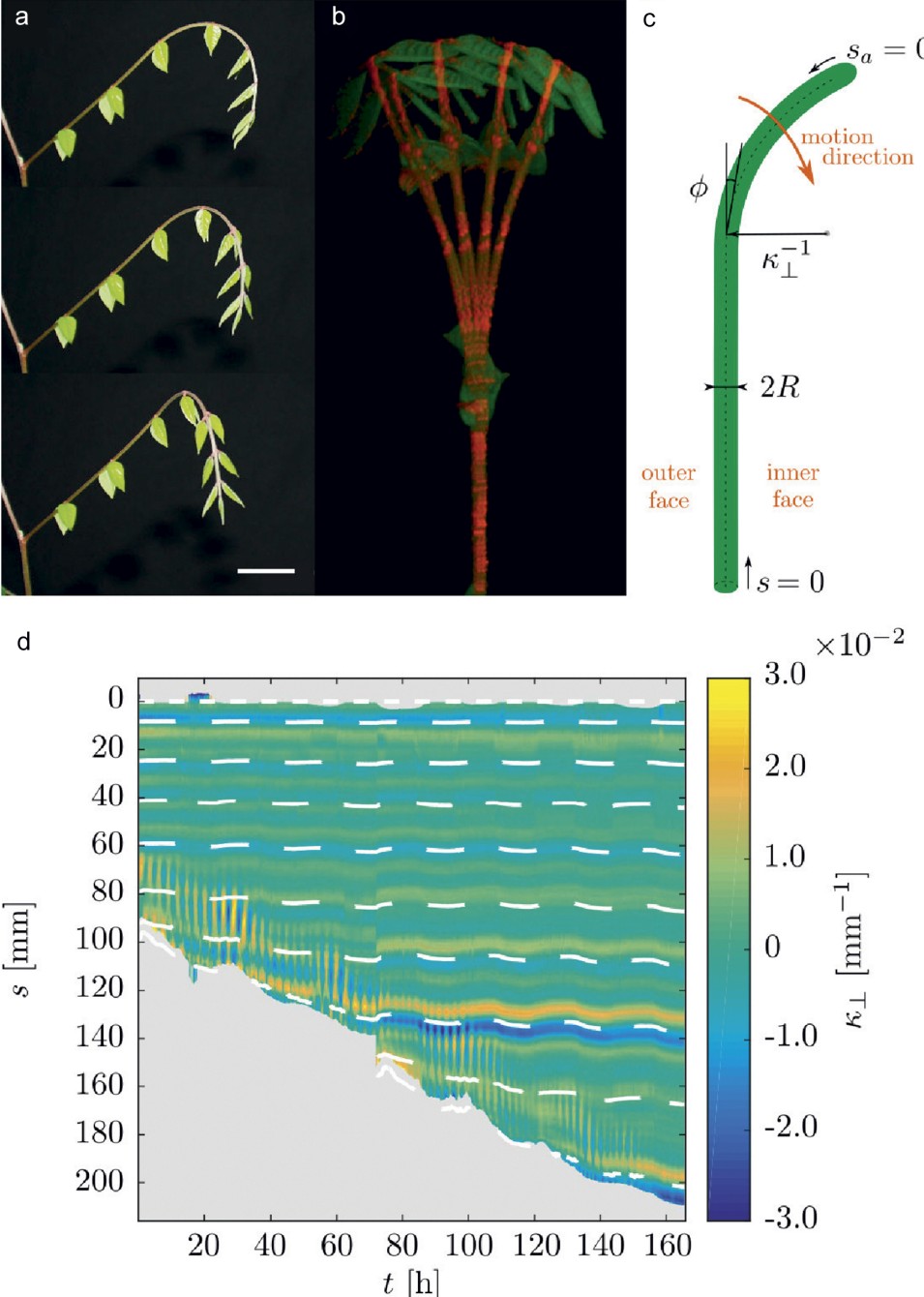

**Figure 1.** Nutation movement of an *Averrhoa carambola* compound leaf. (a) Side view, 30 minutes between pictures from top to bottom. The hook shape gradually comes out of the plane towards the observer. The scale bar indicates 3 cm which is the typical length scale between mature leaflets (b) Top view, 15 min between pictures (nutation period usually varies between 1.5 and 4 hours). The distal end of the leaf oscillates in a pendulum-like fashion, orthogonal to the rachis' axis. After a full period, the leaf has elongated. (c) Geometrical parameters describing the rachis and nutation: arclengths $s$ and $s_a$ (from the base or the apex, respectively), local angle $\phi$, local curvature $\kappa_\perp$ and radius $R$. The direction of motion defines the outer and inner faces of the rachis. (d) Spatiotemporal diagram of the curvature $\kappa_\perp(s,t)$ along the rachis obtained from a top-view time lapse movie. Oscillations of $\kappa_\perp(s,t)$ are visible close to the apex. Dashed white lines mark the position of leaflets.

wavelet transform toolbox. We used the 'cgau2' mother wavelet (second-order derivative of the complex Gaussian). For each location of the rachis, $\dot{\varepsilon}(t)$ was wavelet-transformed. From the resulting complex coefficients $C_{a,b}$ we extracted information on the weight of each scale/frequency in the signal by computing an 'energy': $E(a) = \sum_b |C_{a,b}|^2 / \sum_a \sum_b |C_{a,b}|^2$, where $a$ and $b$ are the scale and shift parameters of the wavelet transform. This information was then re-aggregated and re-arranged to build kymographs displaying the weight of frequencies in the elongation signal along the rachis.

### 2.6. Kinematic model of nutation

The rachis is modelled by a two-dimensional beam of width $2R$ (see Figure S1 in the Supplementary Material) and of total length $L_{tot}$. The geometry of the midline is then described with the same

quantities than the actual leaf (see Figure 1c). The model contains only a few essential ingredients:

1. We define the *elongation rate* $\dot{\varepsilon}$ as the relative local growth rate of an element. For example, at arclength $s$, an element of size $\delta s$ as the following local relative growth rate:

$$\dot{\varepsilon}(s) = \frac{1}{\delta s}\frac{d\delta s}{dt}. \qquad (1)$$

The lateral faces of the beam can have different elongation rates $\dot{\varepsilon}_L$ and $\dot{\varepsilon}_R$, giving rise to differential elongation $\dot{\delta}$. We assume that the profile of elongation is linear in the bulk of the rachis:

$$\begin{cases} \dot{\varepsilon} = (\dot{\varepsilon}_R + \dot{\varepsilon}_L)/2 \\ \dot{\delta} = (\dot{\varepsilon}_R - \dot{\varepsilon}_L)/2. \end{cases} \qquad (2)$$

2. An apical growth zone of length $L_{gz}$ of constant length. The elongation rate of the midline $\dot{\varepsilon}$ is thus independent of time and given by

$$\dot{\varepsilon}(s_a) = \frac{\dot{\varepsilon}_0}{2}\left(1 - \tanh\left(\frac{s_a - L_{gz}}{\Delta L}\right)\right), \qquad (3)$$

where $s_a$ is the arc length starting from the apex, and $\Delta L$ the typical length scale of variation of $\dot{\varepsilon}$.

3. Differential elongation occurs where the mean elongation rate drops, within a bending zone of length $2\Delta L$ (for justification, see Results). Because nutation is a periodic oscillatory motion, differential elongation is modulated by a sine of period $2\pi/\omega$:

$$\dot{\delta}(s_a, t) = \dot{\delta}_0\left(1 - \tanh^2\left(\frac{s_a - L_{gz}}{\Delta L}\right)\right)\sin\omega t. \qquad (4)$$

4. We assume differential elongation is the unique driver of the bending of the rachis. In our case, since the period of nutation is much smaller than the typical time scale of elongation, we furthermore neglect the advection of curvature. In this case, differential elongation rates ($\dot{\delta}$) and the rate of change of curvature ($d\kappa/dt$) have been shown to be equivalent (Jensen & Forterre, 2022; Silk, 1984). Their relationship is purely geometric and can be simplified in the case $R\kappa_\perp \ll 1$ (for us, $R\kappa_\perp \sim 10^{-2}$). We follow the kinematic calculation provided by Bastien (equation A.43 in Bastien (2010)) with second-order correction in $R\kappa_\perp$ to write

$$\frac{\partial \kappa_\perp}{\partial t} \simeq \frac{1 - R^2\kappa_\perp^2}{R}\dot{\delta}. \qquad (5)$$

Interestingly, equation 5 does not display the dilution of curvature due to average growth. Chavarría showed that the dilution effect is compensated by curvature creation (Chavarría-Krauser, 2006).

The model was implemented numerically with discretised versions of the kinematic equations 3, 4 and 5. When and where $\dot{\varepsilon} < |\dot{\delta}|$, local contractions will occur along the lateral faces of the rachis – i.e., either $\dot{\varepsilon}_R < 0$ or $\dot{\varepsilon}_L < 0$ over a finite spatial extent (see Figure S1 in the Supplementary Material). This depends on the relative values of $\dot{\delta}_0$ and $\dot{\varepsilon}_0$ and the exact threshold depends on the spatial functions chosen to describe $\dot{\varepsilon}$ and $\dot{\delta}$. Here, a sufficient condition for contractions is $\dot{\varepsilon}_0 \leq \dot{\varepsilon}_c = 4\dot{\delta}_0$. Finally, the apparent

elongation $\dot{\varepsilon}_\perp$ observed by a camera is obtained by measuring the orthogonal projection of the simulated rachis onto the plane of observation (see Figure S2 in the Supplementary Material).

## 3. Results

### 3.1. Characterizing nutation

As they grow, *Averrhoa carambola* compound leaves exhibit pronounced growth motions. Putting aside the leaflets, the motion of the rachis can be broken down into two different motions, depending on their plane of occurrence (for anatomical terms, see Figure S3 in the Supplementary Material). The unfurling motion of the rachis of *Averrhoa carambola* mostly takes place in a principal plane (Rivière et al., 2017). The rachis unfolds steadily while propagating a hook shape (Rivière et al., 2020). This hook shape is visible in Figure 1a. This motion is also accompanied by out-of-plane curvature variations. The rachis bends and unbends in a pseudo-periodical way, as if it were oscillating around a rectilinear state. The oscillations can already be seen in Figure 1a. In Figure 1b, we see the same motion from the top and on a slightly longer time range. The period of oscillation varies greatly between 1.5 and 4 hours, typically between 2 and 3 hours, while the typical amplitude is of the order of 25 degrees. Supporting movie 1 in the Supplementary Material shows a time-lapse movie of a typical nutation motion, seen from both sides. To properly describe the nutation motion, we define: the base-to-apex arc length $s$, and $s_a$ its apex-to-base counterpart; $\phi$ the local angle with respect to the average direction of the rachis; and the curvature $\kappa_\perp$ (see Figure 1c). Figure 1d shows the quantification of $\kappa_\perp$ in both time and space.

### 3.2. Elongation and bending are localised

We measured the average elongation rate $\dot{E}$ of each of the successive interfoliolar segments by tracking the position of the successive nodes. The spatiotemporal diagram of $\dot{E}$ shows that only the apicalmost region of the rachis elongates, defining a growth zone near the apex (see Figure 2a).

We then estimated the profile of differential elongation $\dot{\delta}$ along the rachis from the transverse curvature $\kappa_\perp$ measurement, thanks to the several hypotheses described in the Material and Methods section. Its envelope was estimated via a method based on the Hilbert transform (Kincaid, 1966) (for more details, see Supplementary Material). The evolution in time and space of the envelope of $\dot{\delta}$ is displayed in Figure 2b. We see that the differential growth – hence the bending – is spatially limited to a zone downstream of the apex. Similarly to what is done for the elongation, it is thus possible to define a bending zone.

This bending zone is at a roughly constant distance from the apex, similarly to the constant length of elongation zone from the apex (see 2B). Finally, going a step further in the description of nutation, we notice that the amplitude of the differential elongation – or of the bending – varies in time, reaching a maximum of $3 \times 10^{-2}\ h^{-1}$. These slow amplitude modulations of nutation are, however, not in the scope of the present study.

### 3.3. Differential elongation peaks where elongation drops

Because the growth spatial profile is almost steady in the frame of reference attached to the apex, we can average the measured quantities in time. The averaged quantities $\dot{E}$ and $\dot{D}$ corresponding to mean elongation and differential elongation rates of interfoliolar

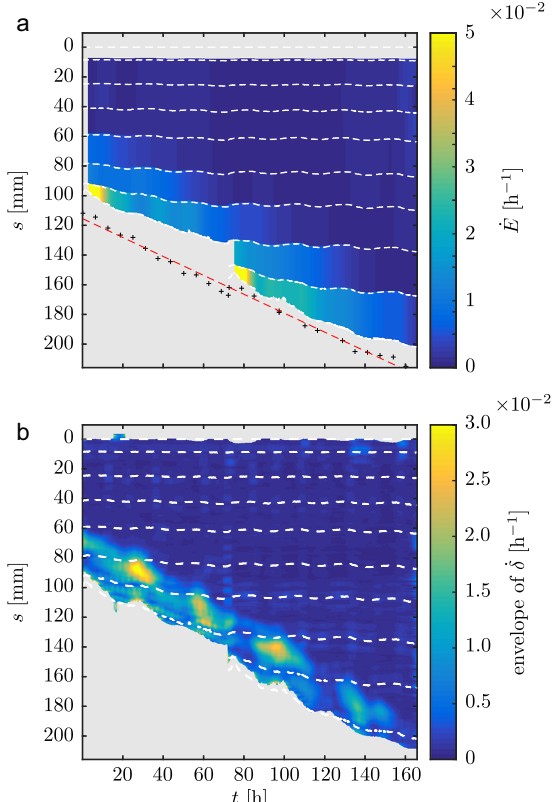

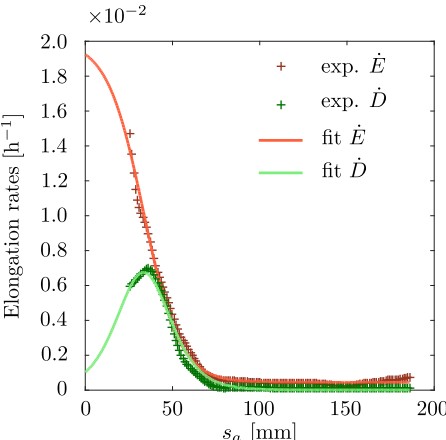

**Figure 3.** Average spatial profiles of elongation rate and differential elongation rate. The two profiles were fitted, respectively, to a sigmoid (red line) and to its derivative (green line). The complete profiles cannot be measured from a top-view because of the hook shape of the leaf.

**Figure 2.** Elongation and estimated differential elongation during nutation. (a) Spatiotemporal diagram of the elongation rate $\dot{E}$ of each interfoliolar segment estimated from the leaflets' trajectories (white dotted lines). The black crosses show the position of the leaf apex estimated from side-view pictures. The red dashed line is a linear fit of the apex position. (b) Spatiotemporal diagram of the envelope of differential elongation $\dot{\delta}$ estimated from the curvature diagram (nutation amplitude).

segments are plotted in Figure 3. Both profiles confirm the existence of a localised growth zone. The typical length scale is about 50 mm, and beyond 100 mm growth is not detectable at all. The mean elongation rate looks like a sigmoid function. In the growth zone the typical elongation rate is of the order of $10^{-2}\text{h}^{-1}$, consistently to typical averaged values found in the literature (Lambers & Poorter, 1992; Poorter & Remkes, 1990), and then decays to zero. Interestingly, the differential elongation rate behaves differently. It is non-monotonic and its maximum coincides with the edge of the growing zone, where the mean elongation rate drops. A simple mathematical description of these sigmoid and peaked shapes is well fitted with the hyperbolic functions as in equations 3 and 4. The results are displayed in Figure 3. In this case the derivative of the fit of the longitudinal elongation rate matches well our experimental measurements of the differential elongation rate, with its amplitude remaining a free parameter (see Supplementary Material).

### 3.4. The elongation profile in the growth zone is compatible with local contractions

We used techniques inspired from digital image correlation (see Materials and Methods) to quantify the elongation profile within the bending zone. However, as the nutation moves the rachis towards or away from the camera, we can only measure an *apparent* elongation rate $\dot{\varepsilon}_\perp$ (see Figure S4 in the Supplementary Material and associated text). Strong projection artifacts indeed affect our

measurements: we see oscillations and even negative values of $\dot{\varepsilon}_\perp$ (see Figure 4a).

Strikingly, the period of oscillation depends on position (see Figure 4a). Oscillations are faster at the apical end of the sample (top on graph), and slower at its basal end (bottom on graph). A wavelet transform evidences two distinct dominant modes with periods in a 2:1 ratio (see Figure 4b). We measured $\tau_f \approx 2.1\text{h}$ at the basal end – corresponding to the nutation period – and $\tau_{2f} \approx 1.2\text{h}$ at the apical end. In an attempt to rationalise these artifacts, and to work around them, we built a simple model based on the experimental kinematic features of nutation and also accounting for projection effects (see Materials and Methods). This model first provides an order of magnitude for differential growth. Indeed, it can be shown that

$$\Delta\phi = 2\Delta L \frac{\dot{\delta}_0}{\omega R}. \tag{6}$$

This can be understood as $\dot{\delta}_0/\omega$ being the total differential growth over one period of nutation, which divided by the radius $R$ gives the local curvature of the rachis, and integrated over the bending zone length $2\Delta L$, gives the final deviation of the apex (see Supplementary Material for formal derivation). By injecting estimations in this relationship ($\Delta\Phi \sim \pi/6$, $2\pi/\omega \sim 2$ h, $R \sim 0.25$ mm and $\Delta L \sim 50$ mm), we find $\dot{\delta}_0 \sim 7.5 \times 10^{-3}\text{h}^{-1} \sim 10^{-2}\text{h}^{-1}$ matching the order of magnitude of the measured average growth, thus confirming the possibility of contractions.

Second, simulations of our model reproduce the observed pattern of $\dot{\varepsilon}_\perp$ (see Figure 4c,d). Our model indeed shows that the two main oscillating contributions to $\dot{\varepsilon}_\perp$ are brought by: (i) projection (geometrical) effects, with frequency double that of nutation, maximum at the apical end of the rachis; and (ii) the differential elongation itself, with frequency equal to that of nutation, peaking around $s_a = L_{gz}$ (see Supplementary Material for more details). While oscillations of $\dot{\varepsilon}_\perp$ at $\tau_{2f}$ are expected in any case (see Figure S4 in the Supplementary Material), oscillations with period $\tau_f$ are a direct signature of differential elongation.

Finally, we fit the wavelet transform spatiotemporal diagram as a way to estimate the unknown experimental parameters. The best fit is presented in Figure 4c,d. The corresponding parameters $\dot{\delta}_0 = 4.5 \times 10^{-3}\text{h}^{-1}$ and $\dot{\varepsilon}_0 = 1.4 \times 10^{-2}h^{-1}$ indicate that the rachis *must* locally contract to explain our experimental measurements.

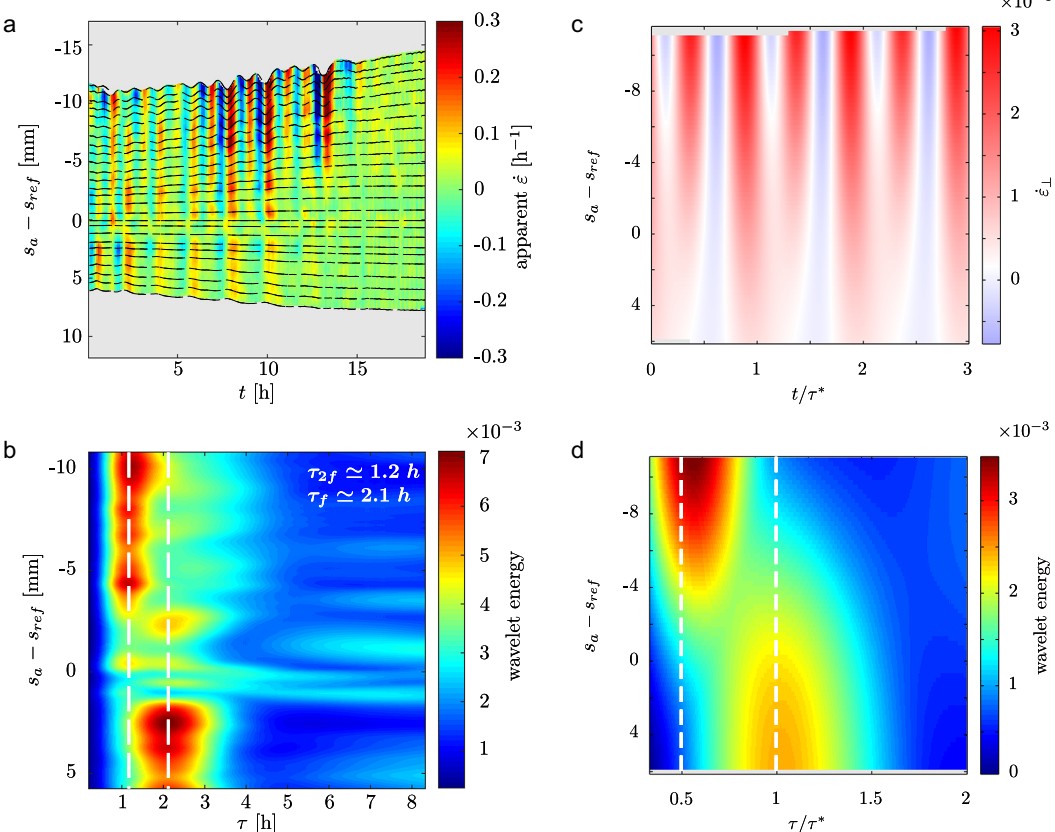

**Figure 4.** (a) Spatiotemporal diagram showing an experimental measurement of the *apparent* local elongation rate $\dot{\varepsilon}$ in the bending zone from a side-view time lapse movie. Because of the oscillatory motion of the rachis, the elongation rate measured is affected by projection effects. (b) Wavelet decomposition of the experimental spatiotemporal diagram of apparent elongation rate. The decomposition shows that two dominant modes in the signal: $\tau_{2f} \approx 1.2$h and $\tau_f \approx 2.1$h, respectively, close to the apical and basal ends of the observed section of the rachis. (c) and (d): Best fit of the kinematics model to the experimental data; $\Delta\phi = 8°$, $L_{gz} = 20.6$ mm, $\Delta L = 12.2$mm, $\dot{\delta}_0 = 4.5 \times 10^{-3}$h$^{-1}$ ($\dot{\varepsilon}_0 = 1.4 \times 10^{-2}$ h$^{-1}$, R = 0.26 mm were measured and fixed before fitting). This set of parameters allows local contractions.

## 4. Discussion

### 4.1. The nutation zone is spatially linked to the growing zone and undergoes 'stop and go' phenomena

The kinematics of nutation presented here are consistent with our previous study on the same system and confirm the presence of a steady growth zone, extending from the apex over a constant length (Rivière et al., 2020). This is also in agreement with growth spatial profiles observed in roots (Chavarría-Krauser et al., 2008; Quiros et al., 2022; Silk et al., 1989; Walter et al., 2002), and several cylindrical aerial organs (Bastien et al., 2018; Peters & Tomos, 2000; Silk, 1992).

We also show that the basal end of the growth zone coincides with the nutation zone – i.e., fluctuations of the differential elongation rate. The spatial coincidence of the maximum of the differential elongation rate with the region of steepest decrease of the average elongation rate is consistent with previous observations on *Arabidopsis thaliana* roots (Chavarría-Krauser et al., 2008). This phenomenon could be compatible with the existence of a maximum value for the elongation rate, likely set by a combination of environmental factors and inner physiological constraints. Close to the apex, growth-regulating signals could be so strong that the elongation saturates by far. Small perturbations of these signals in space or time would not affect the saturated elongation rate and would get edged out. Conversely, when and where they are not strong enough to saturate elongation anymore, any perturbation on

the growth-regulating signals could directly affect the elongation rate and would eventually translate into oscillations. The basal end of the growth zone would then be the location most prone to such variations. The same interpretation could apply to oscillations during the gravitropic straightening of wheat coleoptiles (Bastien et al., 2018): as the coleoptile bends towards the vertical, the differential growth signal is at its maximum, and no oscillation is observed. On the contrary, when the coleoptile approaches a vertical posture, the signal decreases, and nutation of the tip becomes visible again.

Quantitatively, when and where the differential elongation rate is maximum, its amplitude is also comparable to the local average elongation rate (see Figure 3) making the total growth of one side close to zero or even possibly negative. This could be schematized as a 'stop and go' phenomenon, where each side of the rachis grows alternately, before growth and motions cease altogether. This alternate growth behavior was already apparent in pea's epicotyls observation (Baskin, 1986).

### 4.2. Contraction events during plant growth

In all generality, the spatial arrangement of the average elongation rate $\dot{\varepsilon}$ and the differential elongation rate $\dot{\delta}$ can lead to local contractions within the bending zone depending on their relative amplitudes (see Figure 3d). Our local measurements of $\dot{\varepsilon}$ in the bending zone (see Figure 4a,b), interpreted by taking projection effects into account, indirectly revealed that nutation in *Averrhoa*

*carambola* rachis is compatible with local contraction events – i.e., negative elongation rates over finite spatial extent – (see Figure 4). These results are in line with previous reports of contraction events in the circumnutating stems of several other species (Baskin, 1986; Berg & Peacock, 1992; Caré et al., 1998; Stolarz et al., 2008), both at the cell and tissue levels. It was also observed that contractions are circumscribed to either the basal end of the growth zone – where the average elongation rate decays – (Berg & Peacock, 1992), or to the bending zone (Caré et al., 1998), consistently with our findings.

Reports of contractions and negative growth rates go beyond the sole context of nutation. They have indeed been observed during shoot apical meristem morphogenesis (Kwiatkowska, 2006; Kwiatkowska & Dumais, 2003; Kwiatkowska & Routier-Kierzkowska, 2009; Long et al., 2020) and the growth of simple leaves (Armon et al., 2021), both at the cellular and organ scales.

The interpretation of negative growth rates is still a matter of debate in the community. In 1992, Berg and Peacock, attributed tissue contractions to a purely elastic behaviour (Berg & Peacock, 1992). In 1998, Care *et al.* showed that tissue contractions were not artifacts but instead due to local cell contraction driven by osmotic changes (Caré et al., 1998). Only recently, theories describing both elasticity and osmotic water fluxes between cells in plants (Cheddadi et al., 2019; Oliveri & Cheddadi, 2025) have shown that effects due to water transport are central in plant morphogenesis: a growing tissue acts as a sink and extracts water from neighbouring cells which acts like a source. In our case, during the nutation movement, the growing side could get water from the opposite side, leading to contractions of the latter.

### 4.3. A window on the physiological implications of nutation and growth

We believe that growth motions, and nutation in particular, offer an experimental framework to probe growth at the microscopic scale. Its oscillatory nature combined with a clear spatial pattern allow to probe a variety of cell wall mechanics, cell wall chemical status and macroscopic growth rates combinations. A full microscopic investigation goes beyond the scope of this article, but we provide in Supplementary Material a set of preliminary experiments constituting a proof of concept.

The first possible experiment is to use our nutating system to probe cell wall elasticity in growing or not growing tissue. Our preliminary experiments seem to indicate a strong correlation between elasticity and growth: the growing side is found softer than the non-growing side (see Figure S4 in the Supplementary Material, and corresponding text). This belongs to a long series of observations correlating growth with changes in cell wall elasticity, by suggesting that growth is faster where the Young's modulus is lower. This phenomenon was evidenced in growing pollen tips (Zerzour et al., 2009), maize roots elongation zone (Abeysekera & McCully, 1994; Kozlova et al., 2019), *Arabidopsis* shoot meristem before primordia formation (Milani et al., 2011; Peaucelle et al., 2011). Similarly, we can probe the changes in chemical status during growth, and our preliminary experiments seem to indicate a change in the methylesterification status of the pectins if the tissue is growing or not (see Figure S5 in the Supplementary Material and associated text).

In our system it is difficult to disentangle the reversible and irreversible contributions to growth as it was done by Proseus *et al.* for the single-cell algae *Chara* (Proseus et al., 1999). It has also been shown in the case of the shoot apical meristem that elastic inhomogeneities (or differences in stress stiffening) could lead to

differential growth (Kierzkowski et al., 2012). Therefore, to discuss the missing link between the observed microscopic properties and the macroscopic contractions, we propose two different hypothetic scenarios.

First, one should consider the reversible processes as they have already been found to be involved in nutation and growth. As mentioned before, Cheddadi *et al.* recently formalised the water fluxes coupling in multicellular organs. They showed in particular that new types of lateral inhibitory mechanisms could amplify growth heterogeneities (Cheddadi et al., 2019): softer tissues are favored to become sinks for water at the expense of the neighbouring cells. In order to investigate this scenario further, one will need to extend the model to incorporate mechanical aspects. Recently, Moulton *et al.* generalised the analytical results of Timoshenko about the growth of 2D bimetallic strips (Timoshenko, 1925) to filaments in 3D (Moulton et al., 2020a). This new framework, which already proved successful to reproduce plant tropism (Moulton et al., 2020b), is an exciting new line of investigation for nutation.

From our preliminary observations, one could also propose a second hypothetical scenario for the temporal events: on the growing side, HG are actively addressed to the cell wall in their native methylated way. Then growth turns to the other side of the rachis following an external or internal signal, and HG are sparsely degraded or recycled by endoglucanase explaining the reduction in staining observed in methylated and demethylated pectins. Here we can indicate that the time scale could be as fast as 30 minutes. Haas et al. (2020) proposed that the expansion part could be solely due to HG filament expansion following the de-methylesterification. In addition, the partial removal of the highly charged polymers following their recycling could as well lead to cell wall compaction in link with the observed tissue contraction.

## 5. Conclusion

To sum up, we provided on a new biological model case (*Averrhoa carambola*), a complete kinematic description of the nutation motion paying especially attention to the 3D effects. Thanks to a kinematic model we could disentangle the projection artifacts, and prove that contractions really happen during nutation. Nutation is found to occur as a steady propagation spatial growth pattern showing co-localisation of the peak of differential growth with the onset of the growing region. Finally, we showed that this macroscopic behaviour can be used as a tool to investigate microscopic properties of the dynamically alternating growing tissues.

**Open peer review.** To view the open peer review materials for this article, please visit http://doi.org/10.1017/qpb.2025.10013.

## Acknowledgements

We thank Elliot Meyerowitz and Raymond Wightman for their support and help on the immunolabelling experiments. We are also grateful to Olivier Hamant and Emmanuel de Langre for continuous feedback on our works.

**Competing interests.** The authors declare no competing interests exist.

**Data availability statement.** The datasets generated during and/or analysed during the current study are available in the Zenodo repository at https://doi.org/10.5281/zenodo.7994913.

**Author contributions.** All authors designed the research. M.R. and J.D. performed kinematics experiments. M.R. and A.P. performed microscopic

investigation. M.R. and S.D. designed the model. M.R. performed numerical simulations. M.R. and J.D. analysed the data. All authors discussed the results. M.R. and J.D. drafted the article. All authors edited the manuscript.

**Funding statement.** This research received no specific grant from any funding agency, commercial or not-for-profit sectors. M.R. is grateful to 'Ecole Doctorale Frontières du Vivant - Programme Bettencourt' for financial support. IJPB benefits from the support of Saclay Plant Sciences-SPS (ANR-17-EUR-0007).

**Supplementary material.** The supplementary material for this article can be found at https://doi.org/10.1017/qpb.2025.10013.

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
