## [Reviewer Report]

This paper by Rivière and colleagues investigates the spatio-temporal kinematics of nutation in Carambola leaves. In particular, they show that the nutation and elongation occur through a growth pattern that propagates like a traveling wave. The authors suggest that contraction may occur in the shoot, a phenomenon that has been rarely reported in plants. Generally, a better understanding of the bulk mechanics and kinematics of shoots during tropism and nutation—beyond simple curvature measurements—is timely. Although the present study is mainly kinematic, the authors provide insight into potential mechanisms sustaining nutation, in particular periodic changes in elastic properties (however, the authors admit that their results are rather preliminary).

The paper is overall well written and pleasant to read and should appeal to the general readership of QPB (despite numerous typos and problems with the referencing of figures). I have listed my comments below.

Major comments

Introduction

p2–L3: “autonomic or paratonic” – Please define briefly.

p2–L25: “The viscoplastic framework formalized by Lockhart” – Technically, Lockhart uses a constitutive law that is equivalent to that of a Bingham plastic (without any viscosity involved). More generally, authors have argued that growth and plasticity are distinct processes, which are mechanistically and thermodynamically non-equivalent. Thus, the two terms should probably not be interchanged (see for instance Goriely 2017, §4.4.5). In plants, the expansion of cell walls is a complex process that involves rupture of cross-links, remodeling, and turnover—a process very different from traditional plasticity (e.g., in metals).

p2–L41: “contractions in the sunflower hypocotyl” – Is this a contraction of the bulk or an overall shortening of the stem?

Material and Methods

p5–L10: “Differential elongation occurs where the mean elongation rate drops, within a bending zone of length 2ΔL” – What motivates this modeling assumption?

Equation (4): It is not clear to me where this expression originates. Could you provide more details? In particular, there seems to be a missing term. I understand that the authors aim to neglect advection, but in a growing curve, there is also a dilution of curvature. Consider a curve with ẟ₀ = 0 but a constant ϵ̇ ≠ 0 (for instance, a growing circle). In this case, the curve expands isotropically, and its curvature decreases as κ̇ = - ϵ̇κ. Is this term also neglected?

It would be helpful if the authors began with a geometrically exact relation, from which simplifications are explicitly introduced.

Additionally, the model appears to assume the absence of internal stresses between the two sides, as the deformation is explicitly prescribed and linear. As a result, the model (as presented) is purely kinematic and does not incorporate mechanical aspects.

The problem of differential growth within a bimetallic strip was solved exactly a century ago by Timoshenko (1925) and later generalized to filament growth in 3D (Moulton et al., 2020a), particularly in the context of plant tropisms (Moulton et al., 2020b). These approaches include a derivation of Eq. (4) from mechanical equilibrium principles via dimensional reduction. Perhaps these references could be incorporated into the discussion to explore potential mechanistic aspects.

Results

p7–L25: “We then retrieved its envelope thanks to a Hilbert transform” – Please provide details in the appendix.

Discussion

p14–L30: “An unstability favors the softer tissues” – I am not sure if we can talk about an instability here. As far as I remember from Cheddadi and colleagues' paper (correct me if I am wrong), fluxes induce an amplification mechanism that increases growth heterogeneity compared to a scenario without fluxes. However, I don’t think the authors have shown that this mechanism generates instabilities in a mathematical sense.

Supplementary Materials

Figure S6: Please remind the reader of the number of sampling points. Also, given the small amount of data, I am not sure that the observed standard deviation is the best way to show variability. I’d rather opt for confidence intervals instead.

Typographical Errors and Other Minor Issues

Title: “Plant nutation relies on steady propagation of spatial asymmetric growth pattern” – Should this be “spatially asymmetric growth pattern”?

p1–L10: “To whom correspondance should be addressed” – “correspondence”

p2–L36: “an hydromechanical” – “a hydromechanical”

p5–L12: “sinus” – “sine”

p5–L16: “of the kinematic” – “of the kinematic equations”

Figure 1: Is there any particular reason why the plant (and the y-axis) are upside down on the kymograph?

Figure 2: “Elogantion” – “elongation”

p9–L6: “after 100 mm” – “beyond 100 mm” / “past 100 mm”

p9–L8: The “h” of “hour” should not be italicized.

p9–L5: “are plotted on Fig. 2D” – Do the authors mean Fig. 3?

p9–L10: “non-monotonous” – “non-monotonic”

p9–L13: “The results are displayed Fig. 2D.” – Same issue as previously.

p11–L10: “see Fig. 3C-H” – There seem to be issues with the referencing of figures. Fig. 3C-H doesn’t exist. Likewise, Fig. 4 is not referenced, to my knowledge. Some figures are misreferenced in the supplementary materials (shown as ??). Please check carefully.

p13–L34: “the growing side could get water from the opposite side, leading to contractions” – Should this be “leading to contraction of the latter”?

p14–L1: “An open window” – Should this be “A window”?

p14–L27: “mentionned” – “mentioned”

p14–L30: “unstability” – “instability”

p15–L11: “kinematics model” – “kinematic model”

References

[Gor17] Alain Goriely. The mathematics and mechanics of biological growth, volume 45 of Interdisciplinary Applied Mathematics. Springer-Verlag, New York, 1st edition, 2017.

[MLG20] Derek E. Moulton, Thomas Lessinnes, and Alain Goriely. Morphoelastic rods III: Differential growth and curvature generation in elastic filaments. Journal of the Mechanics and Physics of Solids, 142:104022, 2020.

[MOG20] Derek E. Moulton, Hadrien Oliveri, and Alain Goriely. Multiscale integration of environmental stimuli in plant tropism produces complex behaviors. Proceedings of the National Academy of Sciences of the United States of America, 117(51):32226–32237, 2020.

[Tim25] Stephen Timoshenko. Analysis of bi-metal thermostats. Journal of the Optical Society of America, 11(3):233–255, 1925.

---

## [Reviewer Report]

The manuscript presents an original experimental and theoretical study on the nutation of Averhoa carambola plants. The observation that nutation originates from differential growth located only in the transition zone between the growth zone and the mature zone is novel and intriguing. This observation calls for new microscopic interpretations of the biology.

The manuscript needs a revision to address some unclear points. The figures called are usually not the one provided, the numbering should be checked carefully.

Here is a detailed list of comments

1. Abstract: it is no not fully clear to me to state : « Our data first reveal that the differential growth driving nutation is localized and peaks where the average growth drops. » I suggest mentioning that there is spatial zone in between the growth zone and mature zone, along the rachis. Since this is the main result, I think it is important to make sure it is understood.

2. Equations 1-3: Explaining that sA the origin of curvilinear abscissa being 0 at apex would help

3. Explain how equation 4 is derived? Is the one used to measure the differential elongation rate?

4. Explain what is meant by local contraction (using \dot\epsilon_R for instance)

5. Typo page 8: K_\perp instead of \kappa_\perp

6. Figure 2A cited at the end of paragraph « Characterizing nutation » seems to have disappeared

7. Fig 2B in text is shown fo figure 2A, Fig 2C in text points to figure 2B

8. Figure 2D seems to call figure 3

9. Figure 3A calls Figure 4A

10. Figure 3C-H and 3I and 3J are not present

11. Discussion: please better explain better that fluctuating of the DIFFERENTIAL elongation are expected at the basal end of the AVERAGE elongation zone

12. Page 13:, line 10 I do not understand the stop and go, could you evidence this alternance of growth, or is it an interpretation?

13. Figure S2: it is not clear what samples n0, n1 and n2 mean

14. A discussion is missing on the period of 2.1h and 1.2h on the different parts. Why is nutation period 2.1h measured at the distal end of the segment ?

---

## [Editor Report]

Dear Professor Derr,

Your manuscript has been fully evaluated by two independent peer reviewers.

Both reviewers highlighted the novelty of your study and appreciated that you investigated the internal/ kinematics of plant tropic movements. They also stressed that some clarification is needed regarding your hypotheses and concepts you are mentioning. They also asked for more details in the way you derived some key equations.

Also, please check figure numbering.

I look forward to receiving your revised manuscript in which you carefully address all points raised by the reviewers.

---

## [Reviewer Report]

I am happy to see that the reviewers treated my concerns seriously and I’m happy to recommend this work for publication. I do have a small comment left:

- Concerning Lockhart’s model. I still think that the term visco-plastic is improper, in the sense that there is really no viscosity in Lockhart’s model. I’d just call it “plastic-like”. But this is not so important.

---

## [Editor Report]

Based on positive feedback by the two original reviewers, I am glad to accept this very good manuscript for publication in QPB. Reviewer #1 still has a minor comment, that you might take into account if you deem it useful.